# Efficacy Improvement of Anomaly Detection by Using Intelligence Sharing Scheme

**Muhammad Tahir** [1,*] , **Mingchu Li** [1,*], **Naeem Ayoub** [2] **and Muhammad Aamir** [3]

1    School of Software Technology, Dalian University of Technology, (DUT), Dalian 116621, China
2    School of Computer Science and Technology, Dalian University of Technology, (DUT), Dalian 116024, China; nomi@mail.dlut.edu.cn
3    College of Computer Science, Sichuan University, Chengdu 610065, China; aamirshaikh86@hotmail.com
*    Correspondence: mtahirshaikh@mail.dlut.edu.cn or muhammad.tahir.shaikh@gmail.com; (M.T.); mingchul@dlut.edu.cn; (M.L.); Tel.: +86-183-4223-1530 (M.T.)

**Abstract:** Computer networks are facing threats of ever-increasing frequency and sophistication. Encryption is becoming the norm in both legitimate and malicious network traffic. Therefore, intrusion detection systems (IDSs) are now required to work efficiently regardless of the encryption. In this study, we propose two new methods to improve the efficacy of the Cisco Cognitive Threat Analytics (CTA) system. In the first method, the efficacy of CTA is improved by sharing of intelligence information across a large number of enterprise networks. In the second method, a four variant-based global reputation model (GRM) is designed by employing an outlier ensemble normalization algorithm in the presence of missing data. Intelligence sharing provides additional information in the intrusion detection process, which is much needed, particularly for analysis of encrypted traffic with inherently low information content. Robustness of the novel outlier ensemble normalization algorithm is also demonstrated. These improvements are measured using both encrypted and non-encrypted network traffic. Results show that the proposed information sharing methods greatly improve the anomaly detection efficacy of malicious network behavior with bad base-line detection efficacy and slightly improve upon the average case.

**Keywords:** intrusion detection systems; global reputation model; anomaly detection; intelligence sharing scheme; CTA-IDS; machine learning

## 1. Introduction

Computer network threats against critical enterprise computing infrastructure have become increasingly sophisticated. Targeted attacks are on the rise and cybercriminals are launching their campaigns through a variety of legitimate vectors, such as the web or email, which are harder to identify [1]. Therefore, intrusion detection systems (IDSs) are designed to detect attacks that successfully breach the perimeter, after policy enforcement systems, such as firewalls, proxies, intrusion presentation systems, and other perimeter devices [2]. Recently, an increase in both targeted and automated attacks, and well-funded or even state-level adversaries, have been observed. To mitigate these threats, sophisticated and layered defense mechanisms are needed. The first line of defense is typically deployed on the network perimeter: a boundary of a network that is composed of various Internet-facing devices, including boundary routers and firewalls. These are configured to block any easily-identifiable illegitimate traffic [3]. The second line of defense is an IDS: a system built to analyze events within the network for any evidence of ongoing malicious activities. Such evidence is reported to the network administrators so they can react accordingly [4]. IDSs can operate on various levels

of computer system architecture and are classified into two broad categories (host-based IDS and network-based IDS), as discussed below:

Host-based IDS monitors events occurring within a single host for suspicious activity. Characteristics such as network activity, system logs, running processes or application activity can be measured. A typical example of a host-based IDS is antivirus software preforming static analysis. Host-based IDSs are most commonly deployed on publicly available machines containing sensitive information.

Network-based IDS monitors network traffic to detect malicious traffic flows, such as denial-of-service (DOS), distributed denial-of-service (DDoS) attacks, and traffic generated by malicious software (malware) on hosts on the network. Traffic within organization's network can be monitored, as well as traffic to and from external networks (e.g., the Internet).

Previously, different methodologies have been used by network-based IDSs which can be categorized by the level of inspection (deep packet inspection and shallow packet inspection) and method of detection (signature based and anomaly based). These different categories have different characteristics as discussed below:

Deep packet inspection exploits the data contained in packet payloads and operates on the application layer of the network stack. Application-specific detection techniques can be deployed targeting, for example, JavaScript code in web pages [5], or domain name system (DNS) with NX-Domain responses [6]. With encryption becoming the norm [7], deep packet inspection is becoming non-effective since the packet payloads are inaccessible. Encrypted traffic can either be disregarded, lowering the recall, or a man-in-the-middle attack can be performed, which may not be possible or desirable in various circumstances. Moreover, the sheer volume of transferred data often makes these techniques impractical due to unavailability of enough resources to deeply analyze a significant portion of network traffic.

Shallow packet inspection exploits only information in packet headers and meta-information such as request-response delay. This method is less granular than deep packet inspection, but it can be used on larger volumes of data and its function is not impacted as much by encryption.

Signature-based detection systems search network traffic for predefined signatures indicating malicious behavior. Their advantages are low false positive and false negative rates and a comparatively simple design. However, they rely on expert-created signatures that are expensive and time-consuming to create, and can respond to new threats only with a non-trivial delay. While used extensively in the past, they are becoming less relevant due to increases in threat variability, pervasive use of encryption and other evasive techniques.

Anomaly-based detection systems rely on anomaly detection, which is the problem of finding patterns in data called "outliers or anomalies" that do not conform to expected behavior [8]. The major benefit of anomaly-based methods is the ability to detect previously unknown threats or whole classes of attacks based on broad behavioral characteristics rather than precise signatures. This often comes at the expense of higher false-positive rates and greater complexity. High false-positive rates cause a need for manual sifting through the data to extract actionable knowledge, which limits the utility of anomaly-based IDSs in practice. In this scheme, we focus on improving an anomaly-based IDS. The major challenges and characteristics of such systems are described in the following subsection.

*1.1. Major Challenges in an Anomaly-Based IDS*

There are several major challenges that all anomaly-based IDSs face [9]. It is argued in [10] that due to these challenges, network anomaly detection is a fundamentally harder problem than classical machine-learning tasks such as classification. These major challenges heavily influence the design decisions for such system. In order to understand the design and evaluation decisions, they are discussed as follows.

- Huge volume of data: Network security is clearly a big data problem. Network hosts generate lots of data every day. Essentially, one terabyte of data can be easily transferred over the perimeter of an enterprise network in one day. Such a large volume practically prevents deep analysis of all the data, such as deep packet inspection. Many researchers therefore focus on the analysis of an aggregation of information of the network communication in the form of network traffic logs. Nonetheless, the number of log records can easily exceed 10 billion per day, which requires detection techniques to have very low computational space complexity [11].

- Outlier detection: Network anomaly detection is an outlier detection problem. However, machine learning algorithms excel much more at finding similarities and patterns rather than data that does not conform to those patterns. This problem can be side-stepped by viewing anomaly detection as a classification problem with two categories: background (legitimate) and anomaly (malicious). To learn efficiently, machine learning algorithms typically need examples of all categories, preferably comprehensively capturing each category's characteristics. We can learn using known malicious samples, but this leads to a scenario where our algorithm is well suited to finding variants of known behavior, not novel malicious activity [8]. If it is desirable to capture novel threats then, by definition, training samples of only one class are available. This leads to the situation where this study is trying to find a class based on characteristics. This class does not exist in malicious samples, which is certainly not ideal. In order to do that, an accurate model of background behavior is required, which is challenge to provide the large diversity of network traffic.

- The diversity of network traffic: Due to the fact that the Internet is a universal system for communication, there is an immense number of applications such as sites, services and users, all contributing to the diversity of network traffic. When network behavior is marked as anomalous, it may be because it is a rare form of behavior observed for the first time instead of it being malicious. This can lead to high false alarm rates of IDSs, limiting their utility in practice. The impacts of this problem are exacerbated by class imbalance and the resulting high false-positive costs discussed below.

- Class imbalance: The amount of background data by definition is much greater than the number of anomalies and the difference is often many orders of magnitude. In order to have acceptable precision due to the base-rate bias, the false positive rate of an anomaly detector must be kept extremely low [12]. Also, due to class imbalance, many classification algorithms are not directly applicable as they may require classes to be of comparable sizes. Smart sampling or cost weighting strategies may need to be employed to overcome this issue.

- High cost of errors: The cost of classification errors is extremely high compared to the cost of errors in other fields of machine learning (character recognition, image classification, etc.). Alarms generated by an IDS are analyzed by human operators and each false alarm costs valuable time and resources. Even a very small rate of false positives can quickly render an IDS unusable [9]. On the other hand, false negatives have a potential to cause serious damage: even a single compromised system can severely undermine the integrity of an enterprise network.

- Data volume: The volume of network traffic that needs to be analyzed is enormous. This alone can make deep inspection of the traffic unfeasible, so only metadata such as connection type and timing are often exploited. Even in that case, the number of records can be large: the IDS developed by Cisco reports more than 10 billion analyzed requests daily [13]. IDSs are frequently built with a complex layered architecture, decreasing with each layer the amount of processed data and increasing with each layer algorithmic complexity.

- Low quality training datasets: This is best illustrated by the fact that a synthetic dataset knowledge discovery and data (KDD) Cup 99 [14] created in the year 1999 is still widely used by the research community as shown in an extensive survey [15]. This is despite its age and serious flaws that have been discovered over time [15]. Recently, Tobi and Duncan presented the problem of KDD 99 datasets [16]. Hindy et al. [17] presented a review of the issues of current datasets and the lack

of the representative attacks. Their main finding results conclude that the benchmark datasets lack real world properties and fail to cope with constant changes in attacks and network architectures. Also, high-quality training datasets for IDS development are typically very expensive to construct. Due to the cost, complexity of the task and large data volume, the results may not be very reliable. Another option is to use real-world or simulated background network traffic and add malicious traffic gathered from infected virtual machines. The biggest hurdle to publishing datasets in the security domain is privacy concerns. Data may be sanitized by removing or anonymizing potentially sensitive information.

### 1.2. Main Goals and Contributions

There are several ways in which global intelligence sharing could improve detection capabilities of participating networks. There are networks that use man-in-the-middle (MITM) attacks to inspect hyper text transfer protocol secure (HTTP(S)) connections and provide the full set of features for the individual hypertext transfer protocol (HTTP) requests that were sent through an HTTP(S) tunnel. Intelligence gathered from those networks can be used to improve detection efficacy on networks that do not use MITM inspection.

- Some false positives might be specific to a single network and can be removed when using global intelligence. For example, there might be flows that are labeled as anomalous in the context of one network, but are labeled as normal in other networks, which would suggest that it is a case of a false positive. An example of this type of a false positive might be an access to the Yandex search engine [18] that could be easily considered anomalous in American companies.
- The detection efficacy on smaller, geographically-local networks may be poor, because there may not be enough data to meaningfully initialize the system and generate a strong baseline of normal behavior in the individual anomaly detectors. These networks can benefit by importing information from well-initialized networks.

The main goal of this study is to improve the detection efficacy of the existing CTA anomaly-based network IDS (Section 2) on malware that uses network traffic encryption to avoid detection. To achieve this, we utilize information that is currently not utilized by the inter-network data correlations in the CTA-IDS. Although the CTA-IDS is employed on hundreds of different networks, the existing anomaly processing pipeline makes no attempt to share information between individual networks and all the information that is used for anomaly detection is network-local. We propose a novel way of improving the efficacy of anomaly detection by using threat intelligence sharing information gathered from a large number of enterprise networks that are all separately monitored by CTA-IDS. By creating a global intelligence database that is shared among all the participating IDS systems it can be used to improve their anomaly detection performance on both encrypted and non-encrypted network traffic. The major contributions of this study are as follows:

- Build a several variant-based global reputation model (GRM) on top of the CTA-IDS.
- Evaluate the performance of the system variants when used as global reputation lists.
- Use the global intelligence to improve upon existing anomaly detection capabilities in individual networks and measure the results on both HTTP and HTTP(S) traffic.

In this paper, the remaining sections are explained as follows: Section 2 introduces Cisco Cognitive Threat Analytics (CTA) in detail. In Section 3, state-of-the-art work relevant to building a GRM and facing challenges in the application to our domain are investigated by using outlier ensemble models. Several variants of the GRM are also evaluated and compared in this section. The proposed architecture is briefly discussed in Section 4. In Section 5, experimental results of a GRM algorithm are evaluated on real-world network traffic captures and the final results are discussed on both HTTP and HTTP(S) traffic. Section 6 concludes this study along with possible future work.

## 2. Cognitive Threat Analytics IDS

Cognitive Threat Analytics (CTA) [19], developed by Cisco Systems, Inc., is a cloud-based software-as-a-service product designed to detect infections on client machines. It functions as a network-based IDS, analyzing proxy logs produced by web proxies on a network perimeter. It is able to discover data exfiltration, the use of domain generation algorithms, infections by exploit kits, malicious tunneling through HTTP over Transport Layer Security HTTP(S) and command-and-control (C&C) communications [13].

CTA-IDS focuses solely on analyzing traffic using the HTTP and HTTP(S) protocols. The rationale is that when malware communicates with a C&C server by using standard HTTP(S) protocols, it blends the vast amounts of legitimate traffic that is typically generated in any network [20]. Moreover, communication protocols other than HTTP(S) are not universally available as they tend to be filtered out by networks. In the CTA-IDS, the anomaly detection pipeline is a proxy log which is represented as proxy log entries (network flows and simple flows). Each flow contains various fields extracted from the HTTPs headers, such as time of the request, source and destination internet protocol (IP) addresses, HTTP method, downloaded and uploaded bytes, user agent, etc. The full list of HTTP(S) flow fields in CTA-IDS are described in Table 1. The content of the flows differs considerably for HTTP and HTTP(S) traffic. For example, a typical web page triggers many flows of the GET method because various assets need to be downloaded separately. On the other hand, HTTP(S) communication is hidden inside an encrypted tunnel which is typically left open for a time and may contain many HTTP requests. The whole tunnel is represented by a single flow with the method set to CONNECT. This means the encrypted information is unavailable in HTTP(S), such as the user agent string, the resource Uniform Resource Identifier (URI) and meta-information (e.g., request size, request count and timing information).

**Table 1.** HTTP(S) Flow Fields in CTA-IDS.

| Field Name | Field Description | HTTP(S) |
|---|---|---|
| x-timestamp-Unix | timestamp | P |
| x-elapsed-time | elapsed time | P |
| sc-http-status | HTTP status | ∅ |
| sc-bytes | bytes up | P |
| cs-bytes | bytes down | P |
| cs-uri-scheme | URI scheme | Y |
| cs-host | URI host | Y * |
| cs-uri-port | URI port | Y |
| cs-uri-path | URI path | ∅ |
| cs-uri-query | URl query | ∅ |
| cs-username | User name | ∅ |
| s-ip | server IP | Y |
| c-ip | client IP | Y |
| Content-Type | MIME type | ∅ |
| cs-referer | HTTP referer | ∅ |
| cs-method | request method | ∅ |
| cs-user-agent | User-Agent | ∅ |

Table Fields marked with "Y" are available in HTTP(S), fields marked with "P" provide only partial information in HTTP(S) flows due to the fact that many requests may be grouped in one HTTP(S) flow, and the fields marked by "∅" are not available at all. Only 4 out of 17 fields are unaffected by the HTTP(S) protocol Note *: The cs-host flow field is not available in HTTP(S) traffic, but the host name can be extracted from HTTP(S) certificate information.

CTA uses multiple anomaly detectors to assign multiple anomaly values to individual network flows. Anomaly detectors are weak classifier due to the unavailability of cs-host (Table 1). A layered anomaly processing pipeline is employed to combine the anomaly values into a single final per-flow anomaly score. The flows with high anomaly scores are clustered, assigned labels and severities, and finally sorted in the post-processing step. The results are displayed to a network operator so they can react accordingly. The main steps of this processes are defined with further features in the following paragraphs.

Anomaly detectors are used to compute anomaly scores from input flows by using information in a single flow as well as fusing information from the previously-encountered flows and linkage aggregation. There were more than 40 separate anomaly detectors in CTA at the time of analysis of this study. Each is a weak classifier with a real-valued output in the interval [0, 1]; zero is background and one is anomaly. The detectors are designed to be fast, single-purposed, diverse from one another, and to have high recall (sensitivity) at the cost of low precision. Because they are diverse, false positives can later be filtered out in the anomaly processing pipeline, increasing the precision of the output.

The anomaly processing pipeline starts with the anomaly detector outputs and converts them into a single value with high classification strength. This process is illustrated in Figure 1, in which the output of the anomaly detector is processed in three layers. The first layer aggregates the input results of anomaly detectors by using two functions: the unsupervised (Ebtables) function and the supervised (Iptables) function [21]. These functions are designed to increase the precision and preserve recall. The second layer uses Linux-Availability Protection System (LAPS) models [21], to smooth the anomaly values and decrease the false-positive rate. Finally, the outputs from different LAPS models are once again aggregated by GRM and link aggregation bonding functions. The results are averaged together to produce the final anomaly score.

The Linux-Availability Protection System (LAPS) [21] is a method designed to lower false positive rates by smoothing the output of a detector. This way, accidental spikes in anomaly scores are ignored while locally repetitive patterns of malicious behavior are preserved. It works by replacing an output of a detector by a weighted average of itself over similar samples. Similarity is defined as a Gaussian function of Euclidean distance in a predefined feature space [22]. Multiple different feature spaces are used in the pipeline in parallel to achieve better classification strength.

Post-processing of the final anomaly score is the process of converting the numerical output of the anomaly processing pipeline into actionable intelligence that can be presented to a network operator. It involves clustering the flows, attaching labels and severities, sorting by anomaly scores, and displaying the output.

As discussed above, network traffic using both HTTP and HTTP(S) protocols is analyzed, but the amount of information that can be extracted from HTTP(S) traffic is heavily reduced in comparison to HTTP traffic. With many powerful networking companies and organizations advocating encryption and initiatives, such as encryption, decreasing adoption difficulties (for both malicious and legitimate entities), and recent years have witnessed a dramatic increase in encryption pervasiveness. Thus, it is a suitable time to develop threat detection algorithms with excellent performance on encrypted traffic and which improve the performance of existing algorithms.

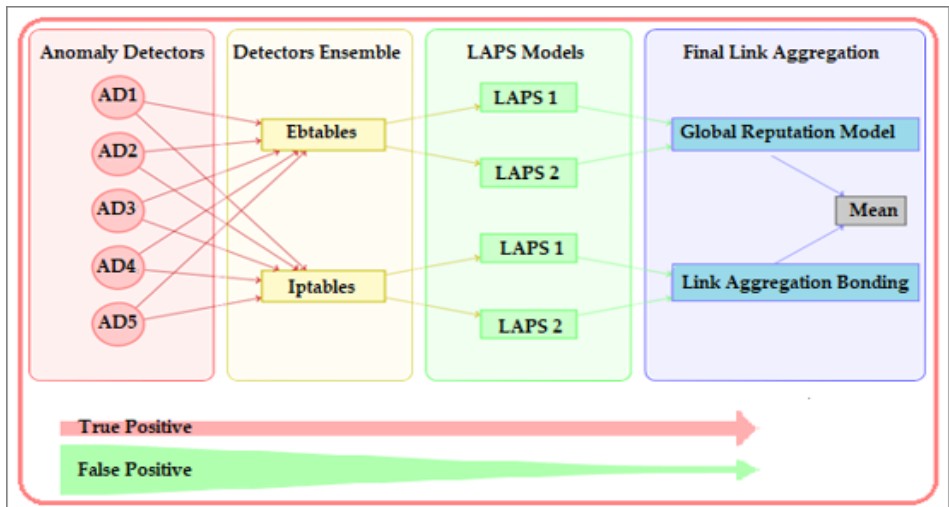

**Figure 1.** Anomaly processing pipeline architecture. (LAPS: Linux-Availability Protection System)

## 3. Related Work and State-of-the-Art

It should be noted that the proposed reputation model is not an instance of a reputation system. Reputation systems let parties rate each other and use aggregated ratings about a given party to derive a trust or reputation score [23]. Notable examples of reputation systems include Google's PageRank [24] or user rating systems on many e-commerce sites. The main challenge faced by reputation systems is that the transitive parties rate each other and potentially malicious parties do not rate each other. Compared to others, this challenge is not faced in the current case because the rating parties (client networks) and rated parties (remote servers) are disjointed.

A better fitting abstraction is the problem of outlier ensemble models. In contrast to better-known classifier ensembles, outlier ensembles are designed to be applicable in an unsupervised setting and their design characteristics are much closer to our needs, as stated in Section 1.1. Individual client networks can be viewed as individual outlier detectors and a global reputation model can be viewed as their ensemble. The output of each outlier detector (client network) is in this case a local reputation model (LRM), associating an outlier score to every observed hostname in each network.

One of the challenges is that individual outlier detectors observe different sets of hostnames. As argued in Section 4.2.1, this property can be treated systematically as an instance of missing data. In the following sections, these ideas are further investigated. By introducing outlier ensemble models, we explore relevant design decisions in outlier ensembles and discuss how they apply to our settings. Finally, the impact of missing values on outlier ensembles is assessed and algorithms dealing with this issue are discussed.

### 3.1. Outlier Ensemble Models

In machine learning, ensemble models are the techniques of combining the outputs of multiple data-mining algorithms to produce a single algorithm [25,26]. Because different algorithms make mistakes in different parts of the input space, this ensemble is often more accurate than any of its constituent parts [27–29]. Ensemble models are widely studied and successfully used in problems such as clustering and classification, but their usage is relatively sparse in the field of outlier detection. Two factors have been major impediments to the success of ensemble models in outlier detection: class imbalance and the unsupervised nature of the problem. Class imbalance refers to the issue that the number of outliers is comparatively small. This makes it difficult to evaluate the success of the algorithm in a statistically robust way and presents a danger of over-fitting. The unsupervised nature refers to the fact that the objective ground truth is often not available to be used to evaluate the quality of components in the ensemble, making it necessary to develop only simple algorithms with few qualitative choices. Nevertheless, outlier ensembles have been treated in the literature a number of times and recent years have seen a number of significant advances in this topic [30].

Aggarwal [31], divides outlier ensemble methods into two categories, as follows.

Model-centered ensembles combine the outlier scores from different algorithms (models) built on the same data set. The major challenge is that the outputs of models are not directly comparable. It may even be the case that high outlier scores correspond to high outlier probabilities in one algorithm and low outlier probabilities in another. Good normalization of the algorithm outputs is crucial for good ensemble performance.

Data-centered ensembles use only a single outlier detection algorithm, but apply it to different subsets or functions of the input data in order to produce an ensemble. Horizontal sampling (a sample of data points) or vertical sampling (a subspace of feature space) can be employed. The earliest formalized outlier ensemble method [32], called "feature bagging" falls into the category of data-centered ensembles, employing vertical sampling of random feature sub-spaces. Because only a single outlier detection algorithm is used, the outlier scores are not nearly as heterogeneous as is the case in model-centered ensembles and normalization does not play such a central role. To improve performance, the sampling procedure can be adjusted as well as the final combination function, and it can still be beneficial to employ normalization as seen in the original feature bagging article [33].

This reputation model can be viewed as a data-centered outlier ensemble, where each ensemble model is a network and data points are the observed hostnames. A typical outlier ensemble meta-algorithm contains three components that are used to arrive at the final result:

1.  Model creation: This is the methodology used to create the individual algorithms (models) that form an ensemble. In data-centered ensembles, it may be, for example, a particular way of sampling input data points or feature dimensions.
2.  Model normalization: Different models may produce outlier scores at different scales. In order for the scores to be comparable, they must be normalized.
3.  Model combination: The normalized outlier scores are combined to form a single outlier score. A weighted average or maximum are common choices of combination functions, since the individual models are already given as individual network outputs. However, the next two steps are necessary components of this reputation model and are analyzed in the following sub-sections.

### 3.1.1. Model Normalization

The simplest approach to model normalization is not to normalize at all. This is possible for some data-centered ensembles where the models do not differ substantially in their output characteristics. In our point of view, this is certainly a viable approach since our ensemble is data-centered and it is unclear how, if at all, the constituent model outputs statistically differ. Their differences may also be meaningful and normalization may in fact decrease the ensemble performance. Normalization has been shown to be beneficial in the case of data-centered ensembles multiple times [28,29]. A good summary of available normalization methods can be found in [34].

Two key properties of outlier scores for the purposes of model normalization are regularity and normality. An outlier score $S$ is called regular if for any scored object $o$, $S(o) \geq 0$, and $S(o) \approx 0$ for an inlier and $S(o) \gg 0$ for an outlier. $S$ is called normal if $S$ is regular and $S(o) \in [0, 1]$ for any $o$. A process of making $S$ regular, respectively normal while preserving the rankings of different data points is called regularization, respectively normalization. Normally, an outlier score is first regularized, then normalized. Normalization can be useful even if $S$ is already normal: the requirements for normality are too lose to guarantee the possibility of meaningful outlier score combination in an ensemble.

Perhaps the simplest regularization method is simple linear scaling. A baseline value $baseS$ is chosen, equal to the expected inlier value. Then, the function $S$ can be regularized as follows:

$$S_{reg}(o) = max\ \{0, \pm s\ (S(o) - baseS)\}$$

The sign ($\pm s$) depends on whether low outlier score $S(o)$ indicates that $o$ is an outlier (plus sign is used) or otherwise (minus sign is used). This regularization method is often used as a preprocessing step to make it easier to apply one of the normalization functions below. A similarly trivial normalization technique would be to scale $S_{reg}$ to fit within the [0, 1] interval in the following way:

$$S_{norm}(o) = \frac{S_{reg}\ (o)}{maX_o S_{reg}\ (o)}$$

Since the maximum function is not a robust statistic, $S_{norm}$ normalization is not robust either, and it is more useful as a preprocessing step to reduce gross calibration differences as a final normalization function.

A somewhat more involved approach to normalization is discarding precise outlier scores and using their ranks only (scaled to fit the range [0, 1]) [28,30]. This neatly solves the normalization issue and makes the outputs of wildly different outlier detection algorithms comparable, but loses a lot of information in the process. The absolute differences between outlier scores are disregarded so there is no way for individual algorithms to express the amount of their confidence in a sample being an outlier or an inlier. Moreover, there is no way to express that an anomalous sample is unavailable at

all. In our case, nearly half of the networks do not contain any anomalies so this makes the rank-only normalization unsuitable.

Another option is to transform outlier scores into probability distributions. Two methods for achieving this goal are described in [29]. The first method assumes that the posterior probability of a sample being an outlier follows a logistic function in relation to its outlier score and learns the parameters of this function from the distribution of outlier scores. The second method models the outlier scores as a mixture of two distributions: exponential for the background and Gaussian for the outliers. Parameters of these distributions are learned and posterior probabilities are calculated using the Bayes' rule. In order to learn in an unsupervised manner, a generalized expectation maximization (EM) algorithm is used to jointly learn the distribution parameters and classifications of data points. Authors show modest improvements over the plain feature bagging in an ensemble model [28]. It favors extreme values of 0 and 1 in the resulting probabilities, which does not lend itself well to combination [30]. Moreover, it is not very stable due to the usage of the EM algorithm and the results can degenerate if there are only a few outliers in a sample, which is frequently the case in the network security field. This means the form of the EM algorithm is not applicable in our case.

Kriegel et al. [30] transforms outlier scores into probability values using a different approach. They assume that the outlier scores follow a chosen distribution to estimate its parameters and use its cumulative distribution function to arrive at the probability values. For example, for a normal distribution, an outlier score $S(\cdot)$ is transformed into a normalized score $Norm1(\cdot)$ as follows:

$$Norm1(o) = max\,\{0, 2.\,cdfgauss(S(o)) - 1\}$$

where $cdfx(.)$ is a cumulative distribution function of an estimated distribution $x$.

Scaling and maximum are used due to interest in a one-tailed probability value only; an average case is assigned the outlier score of zero. Note that the probability $Norm1(\cdot)$ cannot be directly interpreted as a probability of the sample being an outlier; this is not needed since the goal is just making the values comparable. Instead of the Gaussian distribution, other distributions can be used in the same manner; the paper proposes also the use of uniform and gamma distributions. Although it is argued that the precise choice of a distribution is not very important, Gaussian scaling shows the best results in the majority of tests.

In the last step, normalization without assuming a concrete probability distribution is discussed. Instead, the sample means can be normalized to zero and residuals to one by subtracting the sample mean $\mu$ and dividing by sample standard deviation $\sigma$ [35].

$$Norm_2(o) = \frac{S(o) - u}{\sigma}$$

This is similar to the Gaussian normalization $Norm1$ discussed above, but for the use of the $cdfgauss$ function. However, the result is not guaranteed to lie in the interval [0, 1] and high values are not exponentially dampened by the cumulative distribution function that exponentially approaches one.

After being normalized by one of the techniques described above, the ensemble model outputs must be combined to form a single value. In the next section, such model combination functions are described.

### 3.1.2. Model Combination

There are several choices of the model combination function $f: \mathbb{R}^N \rightarrow \mathbb{R}$. The following paragraphs discuss the most common model combination functions.

- Average: It is the obvious choice of a model combination function and has been used extensively in literature [28,30]. It is quite robust to outliers and its estimation is unbiased regardless of the sample size (in contrast to, e.g., maximum). One problem with average has been identified, however: if many models return irrelevant results (for example, when only the minority of

models is expected to assign a high score to an outlier), the average could get diluted. In our model, the diluted average by this mechanism cannot be expected, but the converse may be true: many models could assign high anomaly scores to inliers (background traffic) due to them being observed for the first time. This may be a reason to consider other combination methods, but owning to its robustness, average is the obvious choice.

- Maximum: It is useful for setups where the models in the ensemble display high precision and low recall. It is one of the most common combination functions and it is used particularly in parameter tuning scenarios: the individual models represent the same algorithm over identical data, but with different parameters. The ensemble then picks the strongest result with the maximum function. This choice was discussed in the classical local outlier factor (LOF) paper [36] which implicitly used an outlier ensemble model. Maximum estimation is meaningful for comparison only if all the data points have the same number of components in the ensemble. For our model, this is not true nor do our ensemble models have high precision and low recall, so maximum is not useful for us.

- Minimum: Classifies a point as an outlier only if "all" the models in the ensemble classify it as such. Therefore, it could be useful for models with low precision and high recall. For vertically sampled ensembles this is not the case, because by vertical sampling, recall is diminished. On the other hand, for many outlier detection algorithms, horizontal sampling preserves recall and diminishes precision, so minimum can be a good choice. Horizontal sampling in outlier ensembles is not common so the minimum function is rarely (if at all) used. As is the case for maximum, all data points must have the same number of components in the ensemble, and this excludes minimum from consideration as our combination function of choice.

- Pruned average: It is a way to overcome the diluting effects of a simple average. Low anomaly scores are discarded (either by thresholding or by using only the top *n* scores), and the remaining scores are averaged. The problems with this approach are that it contains parameters that must be tuned (which is difficult to do in an unsupervised setting), it is computationally expensive and some issues of comparability between the scores of different data-points arise. Despite the fact it combines the benefits of average and maximum, it is rarely used in practice [27].

- Damped average: Applies a dampening function such as a square root or logarithm to the values before averaging them. This is used to prevent the result from being dominated by few extreme values. Geometric mean is a special case of damped average, where the dampening function is a logarithm. It does not seem necessary to implement dampening in our case, since the anomaly scores are already normalized and so very extreme values cannot appear.

*3.2. Dealing with Missing Values*

When our reputation model is viewed as a horizontally-sampled ensemble (in Equation (2) below), there are many missing values (Table 4) that require careful treatment.

- Complete-case analysis: Also known as "list-wise deletion". In this method, the entire record is excluded from analysis if a single value is missing. This method is known to behave well even if values are not missing completely at random [37]. In our case, only the hostnames that are found in "all" of the networks would be used for analysis. This is difficult because such a set of hostnames is actually empty. A way to get a non-empty set is to use only a subset of available networks. This leads to a very small resultant dataset which in turn leads to large estimation errors.

- Available-case analysis: Also known as "pairwise deletion". For many models, the parameters of interest can be expressed using population mean, variance, correlation and other population statistics. In available-case analysis, each of these is estimated using all of the available data for each variable or pair of variables. In our case, by assuming the mean and standard deviation estimates are required, the entirety of the available data would be used, but the sets of hostnames

that each network normalization is dependent on would be different. Intuitively, this would give better results than complete-case analysis due to the much bigger dataset available but it also introduces additional bias if the probability of a value being missing is correlated to the anomaly score and it introduces bias in estimating standard deviation due to the different sample sizes.

- Data imputation: Many methods fall into this category; the common trait is that these methods guess or estimate the missing values before doing the analysis. A popular approach is simply to substitute means of existent values for missing values. Another method is to estimate the missing values using the maximum likelihood method or draw the missing values from a chosen distribution multiple times and aggregate the results. If the substituted values are biased, the results of analysis based on them will also be biased. Here, due to the extremely large number of missing values in our case, the data imputation seems not to be the right choice. The methods with strong theoretical guarantees (maximum likelihood and multiple imputation) need precise probabilistic models of the data which are unfortunately not available in our case.

## 4. Proposed Architecture

In order to share intelligence pertaining to the part of observed traffic in a network, a way to identify similar parts of traffic in other networks must be devised. There are two ways to identify similarities in network traffic: remote identity identification and behavioral correlation.

- Remote identity identification: is a simple way to correlate traffic across networks by using hostname or IP address. It can be used in a straight-forward manner to assign the reputation scores to different endpoints or form a blacklist of malicious entities. It is a standard and functional way to enhance network security. Identity identification offers an added benefit of reliably binning captured HTTP(S) traffic together with the corresponding traffic where a MITM attack is used to inspect the contents. On the other hand, pervasive use of identity identification has led to change the originating hosts which is mostly used by attackers to make black-listing less effective.
- Behavioral correlation: is able to overcome such evasive tactics by using the timing, request sizes and request methods, as the resulting features are designed not to be tied to a single identity. Moreover, behavioral correlation techniques exploit a wide range of traffic characteristics and sharing.

Because of its relative merits, identity identification is used as the basis in our intelligence sharing system. There are several ways of identifying the remote parties that are available for network flows such as URI, hostname, IP address and the second-level domain. Out of these, the hostnames can be used as the most specific identity information that is available for both encrypted and non-encrypted flows. IP addresses are potentially more specific because IP addresses are used to serve identical resources for the purpose of load balancing.

### 4.1. Global Reputation Model

Here, the global reputation model (GRM) is proposed, which uses the information gathered from individual networks to assign globally associated anomaly values with individual hostnames. The overall reputation model structure can be visualized in Figure 2, where the input is a list of flows with associated anomaly scores (produced by the anomaly processing pipeline of the CTA-IDS), and the output is a GRM. The GRM can be used to enhance detection strengths in individual networks. There are three parts of this reputation model represented by blue arrows (Figure 2) that are developed and evaluated in this research work.

1.  Aggregation of anomaly scores over hostnames, producing a set of local reputation models (LRMs). This part can be implemented simply as an average of per-flow anomaly values described in Section 4.2.1.

2.  Combination of LRM to form a GRM. This part is much more involved than the others and presents difficult challenges such as normalization of individual local reputation tables. The remainder of this work mainly focuses on the design and implementation of this part of the overall intelligence sharing algorithm.

3.  Improvement of anomaly detection in individual networks using the GRM. This part can be implemented by replacing individual flow anomaly scores with hostname reputation values described in Section 4.3.

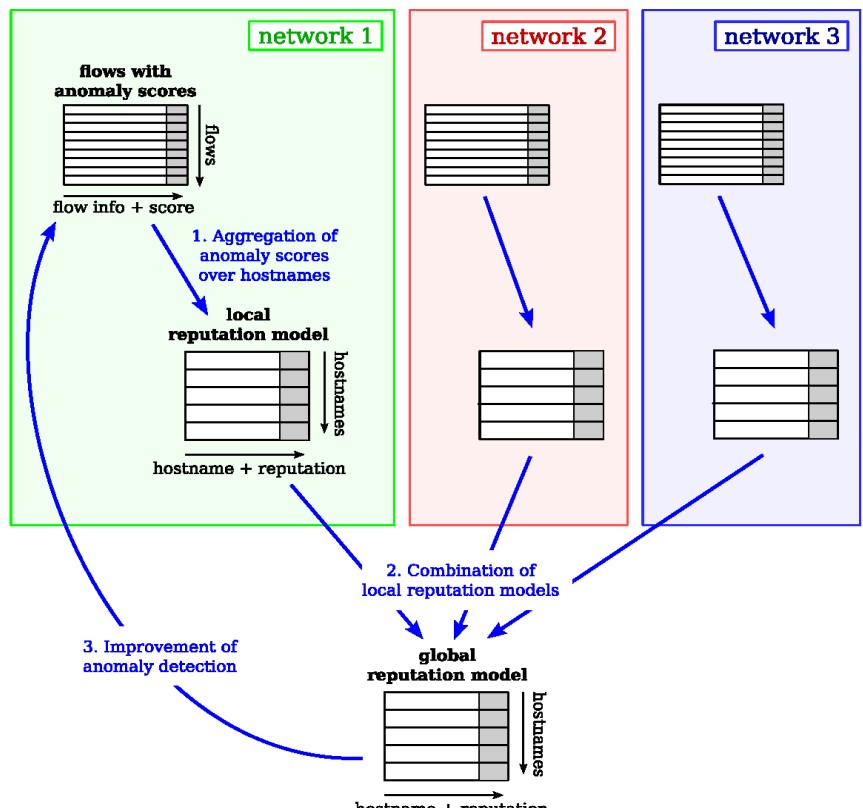

**Figure 2.** Overview of the global reputation model (GRM) illustrated on three networks.

*4.2. Proposed Algorithm*

Several variants of the GRM, based on the considerations in the previous section are proposed here. Initially, we discussed how the reputation model can be viewed as an outlier ensemble model and, then, the implementation choices of the three parts of the ensemble model. Lastly, the properties of a novel normalization algorithm are analyzed that are designed to suffer from missing values less than the state-of-the-art methods.

4.2.1. Global Reputation Model (GRM) as an Outlier Ensemble

In order to use the outlier ensemble abstraction, the transformation of inputs from the form of network flows with anomaly scores into a local reputation model (LRM), is necessary (step 1. in Figure 2). This is represented by treating each flow's anomaly score as an estimate of the corresponding hostname anomaly score. Then the host-name anomaly scores are estimated by averaging the anomaly scores of all flows that communicate with said hostname in a single network. This simple procedure arrives at the local reputation model.

The GRM can be viewed as a data-centered outlier ensemble with horizontal sampling. Let $H$ be the set of all existent hostnames and $Hn \subseteq H$ the set of all hostnames that were observed in the network $n \in N$. Let $fn : Hn \rightarrow \mathbb{R}$ be the anomaly score of a hostname measured in the network $n$. Now, $Hn$ is the horizontal sampling of $H$ and the sets of anomaly scores observed in individual networks form an ensemble $Eh$ :

$$Eh = \{\{fn\,(h) \mid h \in Hn\} \mid n \in N\,\} \tag{1}$$

Horizontally sampled ensembles are routinely used in classification. Classical examples are bootstrap aggregating [38] and random forests [39]. Intuitively, this is understandable; horizontal sampling leads to dilution of dense groups of data points which significantly hampers our ability to identify outliers. To be able to utilize the literature around outlier ensembles, the reputation model is re-defined as an ensemble with vertical sampling by allowing the existence of missing values. The ensemble is now:

$$Ev = \{[fn\,(h) \mid n \in N] \mid h \in H\} \tag{2}$$

In the above equation, $[fn\,(h) \mid n \in N]$ is an ordered set of the anomaly scores of $h$ in all networks. Since not all of the hostnames are observed in every network (the domain of $fn$ is a subset of $H$), there are missing values in the ensemble. The difference between these two representations is illustrated in Figure 3. The figure shows that, in the case of horizontally-sampled ensemble (left), we treat each network as a sampling $Hn$ of a set of hostnames $H$. In the case of the vertically-sampled ensemble (right), each network is treated as a dimension $Dn$ in a feature space of hostnames. Sampling is then conducted over the dimensions of the feature space, one dimension at a time, and values are missing whenever a hostname was not observed in a given network [40].

By considering the reputation model as a vertically-sampled ensemble, explicitly dealing with missing values is necessary. The shortcomings of existing algorithms designed for estimation with missing values is necessary (discussed in Section 3.2) to motivate the design of a novel ensemble normalization algorithm described later in this section.

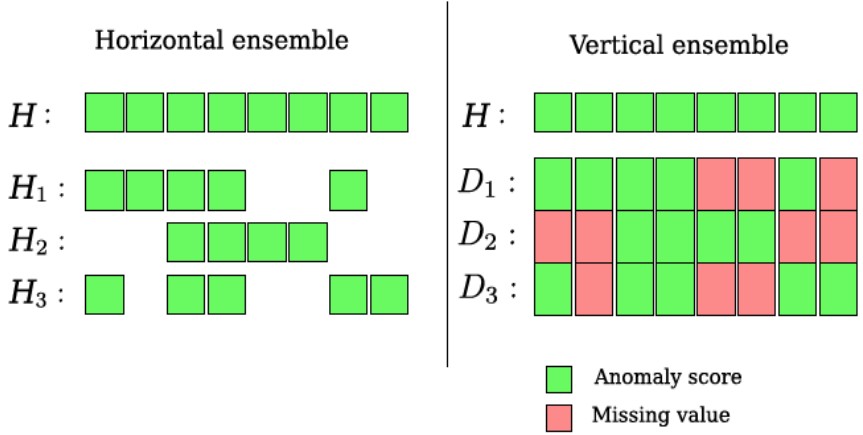

**Figure 3.** Horizontally and vertically sampled ensemble illustration.

### 4.2.2. Normalization

Based on the experiments conducted in [30], the best-performing normalization variant across a wide array of scenarios was Gaussian scaling:

$$Normgauss(o) = max\{0, 2 \cdot cdfgauss(S(o)) - 1\} \tag{3}$$

where *o* is a network flow, *S* is an outlier score, and *cdfgauss* is the cumulative distribution function (CDF) of a Gaussian distribution with parameters estimated from the data.

As can be seen, in Figure 4 the first term of the Taylor expansion of *erf*(*z*). When used as a transformation function, the linear approximation better preserves values far from zero than the *erf*(*z*) function.

This can be rewritten using the Gaussian error function *erf*:

$$Norm_{gauss}(o) = max\left\{0, erf\left(\frac{S(o) - u}{\sqrt{\pi}\sigma}\right)\right\} \tag{4}$$

where $\mu$, and $\sigma$ are the estimated mean and standard deviation, respectively. For our purposes, however, a change to this equation is warranted, as the models in our ensemble are identical (only the data is not) and there is no need for the required result to be inside the [0, 1] interval, by removing the exponential decay towards 1 of the *erf* function.

This decay (illustrated in Figure 4) makes the scores of outliers less pronounced and could arguably lead to their dilution. When *erf* is substituted by its linear approximation around the origin and the maximum function is removed, we get:

$$Norm_{lin}(o) = \frac{2}{\sqrt{\pi}} \cdot \frac{S(o) - u}{\sqrt{\pi}\sigma} \alpha \frac{S(o) - u}{\sigma} \tag{5}$$

This coincides with the standardization procedure which is a frequently used technique in ensemble outlier detection [31]. This normalization function will be used as one choice for our experimental evaluation. The second choice will be the identity function: $Norm_{id}(o) = S(o)$. This will provide a baseline for evaluation of how much (if at all) the results improve when normalization is employed.

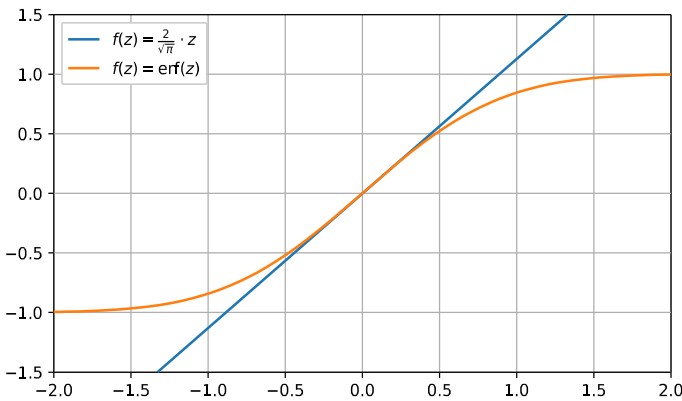

**Figure 4.** Error function decay illustration.

### 4.2.3. Combination

As discussed in Section 3.1.2, only variants of the average function (average, pruned average, damped average) are suitable for our algorithm. Pruned average requires a pruning threshold to be set and damped average requires a dampening function. For neither of these is it obvious how to choose these parameters in an unsupervised setting. Therefore, simple average as our combination function was used, consistent with much of the prior state-of-the-art outlier ensembles.

### 4.2.4. Missing Values

Among the methods useful for dealing with missing values described in Section 3.2, for evaluation, both complete-case analysis and available-case analysis were selected. Because of the large proportion of missing values, data imputation approaches were not pursued. To have enough complete cases for complete-case analysis, only a subset of networks in experiments with this technique were used.

In addition to the standard methods of complete-case and available-case analysis, a novel method was proposed by designing enabled ensemble normalization with higher tolerance to missing values than either of the methods above. This method is represented as "pairwise optimization of normalization error" and is described in detail as follows.

### 4.2.5. Pairwise Optimization of Normalization Error

This method is similar to pairwise deletion (available-case analysis) when used to estimate a correlation matrix. In that case, pairwise deletion was used in pairs of networks $n_i, n_j$ to compute correlation coefficients $c_{i,j}$ and build a correlation matrix $[c_{i,j}]$. For each pair of networks $n_i, n_j$, a set of common hostnames was used to compute the correlation; hostnames missing from some of them were discarded. A much greater proportion of samples was utilized in this way. Then, in the other case with complete-case analysis, only identical sets of hostnames are ever compared together in contrast to available-case analysis. However, the resulting correlation matrix $[c_{i,j}]$ is not guaranteed to be positive-semi-definite.

Since our goal is normalizing outlier scores rather than estimating the correlation matrix, a normalization error function was built instead of a correlation matrix, but it performed in the same pairwise manner. This error function encompasses the statistical differences of all pairs of networks. By minimizing it, the parameters were estimated that minimize the said differences.

### Derivation

Let $N$ be a set of all networks. Each network $n \in N$ contains a set of hostnames, $Hn$. Let $fn : Hn \rightarrow \mathbb{R}$ be an anomaly score of a hostname $h \in Hn$ in network $n$. Let $g\theta n: \mathbb{R} \rightarrow \mathbb{R}$ be a normalization function parameterized by $\theta n$.

We wish to derive the parameters $\theta n$ that normalize best the anomaly scores $fn(\cdot)$. Using these parameters, it is possible to compute the normalized anomaly scores $q_{n,h}$:

$$q_{n,h} = gn\left(fn\left(h\right)\right), \ h \in Hn \tag{6}$$

Now, by using the least squares optimization method to find the best parameters $\theta n$ for every pair of networks $n,m$, the normalization error function $En,m$ is:

$$E_{n,m} = \sum_{h \in Hn \cap Hm} \left(g\theta n \ fn(h)\right) - g\theta m(fm(h)))^2 = \sum_{h \in Hn \cap Hm} \left(q_{n,h} - q_{m,h}\right)^2 \tag{7}$$

Here, for simplicity, we substituted anomaly scores in the normalized anomaly scores $q_{n,h}$. By minimizing $E_{n,m}$ over $\theta n$, $\theta m$, we obtained the normalization parameters $\theta n$, $\theta m$ for networks $n$, $m$. To compute the global error function $E$, we can sum the errors $E_{n,m}$ over all pairs $(n, m)$:

$$E = \sum_{n,m \in N,} E_{n,m} = \sum_{n,m \in N, \ h \in Hn \cap Hm} \left(q_{n,h} - q_{m,h}\right)^2 \tag{8}$$

This can be re-arranged to sum over hostnames in the final results. By doing so, the descriptions of the summation sets would be complicated, so they will no longer print, and we simply assume that a summand is present only if its value is defined. After re-arranging we get:

$$E = \sum_h \sum_n \sum_m \left(q_{n,h} - q_{m,h}\right)^2 \tag{9}$$

where according to the stated convention, $n$ and $m$ are summed only over the networks that observe the hostname $h$. It is now visible that the influence of a single hostname $h$ is quadratic to the number of networks it is present in (there is a quadratic number of summands referencing this hostname). This is caused by using a pairwise error function instead of an error function comparing to a single baseline.

This is certainly not desired; due to the nature of the dataset, it can be expected that the differences between the number of occurrences of different hostnames is large. They would be quadratically exacerbated, and the influence of less frequent host-names would vanish. To alleviate this issue, we introduced a weighting parameter $wh = |\{n\,|\,h \in H_n\}|$ (the number of networks that $h$ is observed in) and used it to weight the error function:

$$E = \sum_h \sum_n \sum_m \frac{1}{w_h} \cdot (q_{n,h} - q_{m,h})^2 \tag{10}$$

The resulting error function $E = E(g\theta_1, g\theta_2, \ldots, g\theta_{|N|})$ contains only sums and multiplications of $g\theta_n$, so it is differentiable with respect to $g\theta_n$, and with respect to $\theta_n$ whenever $g\theta_n$ is differentiable, so it can in principle be optimized by a first-order iterative optimization algorithm.

It should be noted that this error function may not be useful depending on the scale of the error function on parameters $\theta_n$. For example, if we choose mean and scale parameters $g_{un,\sigma n,}(x) = \frac{x - un}{\sigma n}$, the optimal solution $E = 0$ would be for the degenerate case of $\sigma n = 0$, $n \in N$. In cases like this, a more involved error function or a limited domain of parameters would be necessary to get useful solutions. For simplicity, a shift normalization function $g_{un\,(x)\,=x-\,un}$ was used that does not suffer from these problems. This simple normalization function should eliminate the biggest calibration errors and improve the ensemble classification.

After substituting for $q_{n,h}$ and using the shift normalization, the final normalization error function was obtained:

$$E(u) = \sum_h \sum_n \sum_m \frac{1}{w_h} \left( f_{n\,(h)} - f_{m\,(h)} + u_m - u_n \right)^2 \tag{11}$$

After differentiating with respect to $u_n$, the following equation was achieved:

$$\frac{\partial E}{\partial u_n} = \sum_h \sum_m \frac{4}{w_h} \left( f_{m\,(h)} - f_{n\,(h)} + u_n - u_m \right) \tag{12}$$

This gradient function can be used to minimize $E$ by any first-order optimization method, such as gradient descend.

Time Complexity

In the worst-case scenario where all the hostnames $H$ are present in all the networks $N$, the gradient function $\nabla E$ has $|N|$ dimensions and $|H| \cdot |N|$ summands. Combined with the number of iterations $I$ required to get to the minimum, the time complexity $t$ of our algorithm is:

$$t \in O\left( |N|^{2} \cdot |H| \cdot I \right) \tag{13}$$

The $O$ notation reflects the fact that the complexity is linear to the number of hostnames and the number of iterations and quadratic to the number of networks in the ensemble. In practice, this complexity did not pose problems; for approx. 300 networks, 6.5 million unique hostnames and 50 iterations, optimization using gradient descent took approximately 2 min. In our case, the computation was not parallelized, so this solution is very scalable.

*4.3. Reputation Model Usage Scenario*

In this section, a way to use a GRM to improve detection in participating networks is proposed.

Firstly, enough data must be gathered to compute a meaningful GRM. If the time period is too long, the reputation model would suffer from modeling facts that may no longer be valid. If the time period is too short, the estimation of a reputation model could be noisy. The use of three days' worth of network data is proposed and evaluated. After this time period elapses, a GRM is computed and distributed among networks. The networks can then use the reputation model to improve the

subsequent network traffic anomaly detection. Simultaneously, they gather data to build a novel GRM. After any set time period, the newly gathered data can be used to re-build the reputation model and the cycle continues. For simplicity, a once-built reputation model for another three days in experiments was evaluated. It would also be possible to refresh the system much more frequently to ensure maximum freshness of the reputation model.

In effect, the GRM provides each network with a table of per-hostname anomaly scores. These anomaly scores can be used to improve anomaly detection in a network by performing a weighted average with the flow-specific anomaly scores. In proposed experiments, we strive to maximize the impact of the GRM on anomaly detection to achieve maximum contrast between the methods and to be able to better discern any differences between the two. Therefore, for every hostname recorded as malicious traffic data in the reputation model, we replace its anomaly score with the one from the reputation model. Here, no record of malicious traffic data in the reputation model exists, because the original anomaly score of the normalized flow was used accordingly.

## 5. Experimental Evaluation and Discussion

In this section, we evaluate the algorithms proposed in Section 4.2 using real-world network traffic captures. The dataset and methods used for the evaluation of proposed algorithms are described. Finally, the results are also discussed.

Two experiments were performed. The first (global reputation table evaluation) is targeted at evaluating the classification performance of the global reputation models produced by the algorithms proposed in Section 4.2. The second experiment (improvement of per-flow anomaly scores) is aimed at evaluating possible improvements that can be attained by using the global reputation models to strengthen the anomaly scores in individual networks.

### 5.1. Dataset Description

The dataset used for experimental evaluation consists of network communication of client networks that employ Cisco CTA-IDS. Data was gathered over the first week of January 2018. There were in total 622 networks, but only 310 of these networks were chosen at random to produce a sizeable dataset that simulates a realistic use-case while still being reasonably comfortable to work with (541 GB of raw data). Important characteristics of the dataset are shown in Table 2. The dataset entries were labeled by an expert at Cisco as being either legitimate or malicious. These labels are available both at the level of individual flows as well as the level of individual hostnames, and used as ground truth in all experiments. Networks are sorted independently for each line in order to make the lines monotonic. There are 310 companies in total, out of which 143 contain at least one malicious HTTP flow and 97 contain at least one malicious HTTP(S) flow; 6 networks contain zero HTTP flows and 16 contain zero HTTP(S) flows.

Companies vary wildly in the number of flows. To illustrate, 50% of the biggest companies serve 99.0% of the total number of flows. There is a need to pay attention at the time of result analysis. Detailed visualization of the company's identification ID is illustrated in Figure 5.

In order to test the deployment scenario described in Section 4.3, the dataset must be split chronologically into two parts. The first part is used to create and test a global reputation table while the second part is used to evaluate the possible improvements of per-flow anomaly scores using a reputation table. Coupled with the fact that there must be a warm-up period in which the anomaly detectors are learning the characteristics of network traffic and do not provide useful output, the following splits were achieved under the 7 days of data:

- The first day is discarded as a warm-up period.
- The next three days are used to compute the global reputation tables.
- The last three days are used to evaluate the improvements of anomaly score using the global reputation tables.

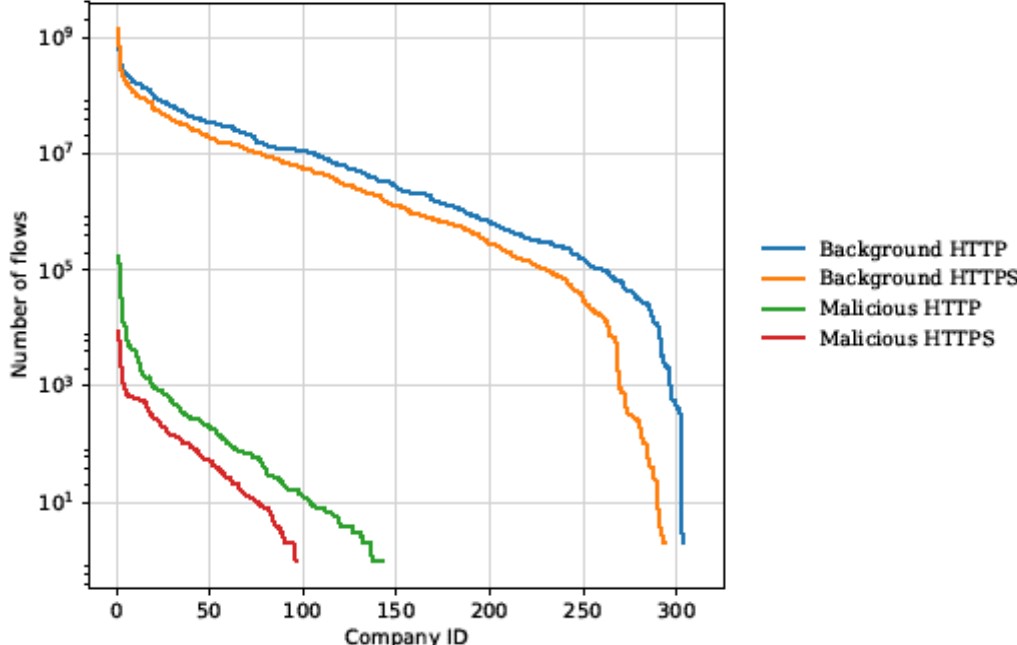

**Figure 5.** Number of flows counts per network according to the number of company's identification ID.

**Table 2.** Characteristics of the datasets used for experimental evaluation.

| Datasets Description Collected Over | Experimental Evaluation Inclusive |
| --- | --- |
| Number of networks | 310 |
| Total number of flows | 12,932,245,944 |
| Total number of malicious flows | 394,951 |
| Proportion of malicious flows | 1:32,744 |
| % of HTTP(S) connects in background flows | 43.2% |
| % of HTTP(S) connects in malicious flows | 7.3% |

*5.2. Evaluation Criteria*

Receiver operating characteristic (ROC) curves and the area under ROC curve (AUC) are used as performance measurement methods. The ROC curve is a graphical plot that illustrates classification characteristics of a classifier as its discrimination threshold is varied. It is created by plotting the true positive rate (TPR) against the false positive rate (FPR) for all possible values of the discrimination threshold. This means any concrete threshold cannot be chosen as the ROC curve captures all possible choices.

The AUC performance measure is simply the area under a ROC curve. It summarizes classifier performance as one number inside the [0, 1] interval, where 1 is perfect classification and 0.5 is equivalent to the performance of random guessing. Another interpretation of AUC is that it is equal to the probability that a classifier would rank a randomly chosen positive sample higher than a randomly chosen negative sample. For convenience, AUC scores are often written as percentages.

*5.3. Global Reputation Table Evaluation*

In this section, the classification performance of four global reputation tables produced by the four algorithms proposed in Section 4.2 are evaluated and compared. All four of the algorithms use average as their combination function; their normalization functions and ways of dealing with missing values differ. The evaluated algorithms are listed below.

1. average: no normalization is performed;
2. norm-isect: $\frac{x-u}{\sigma}$ normalization on the intersection of hostnames (list-wise deletion);
3. norm-all: $\frac{x-u}{\sigma}$ normalization on all observed hostnames (pairwise deletion);
4. pairwise: pairwise optimization of normalization error.

From now on, these algorithms will be referred by their identifications as specified above.

For the norm-isect algorithm, there is need of a meaningful intersection of all networks. This is not the case when all of the networks in the dataset can be used. There are even networks that do not have any network flows in the training portion of the dataset, so the intersection is empty. Therefore, we need to use only a subset of networks for norm-isect to be applicable. A simple way to choose a reasonable set of networks is to choose the *n* networks with the largest numbers of unique hostnames. The sizes of intersections of the biggest *n* networks are visualized in Figure 6. In the given figure, the horizontal axis indicates the number of networks in an intersection (the networks with the largest number of hostnames are taken), and the vertical axis indicates the number of hostnames in the intersection on a logarithmic scale. A total of 100 networks were used, which corresponds to the intersection size of 1054 hostnames. Based on this figure, there is no immediately obvious number of companies that should be chosen. To strike a balance between calibration precision and utility, we arbitrarily choose to use 100 networks; their intersection contains 1054 unique hostnames. The same set of 100 networks is used for all four algorithms in order for their ROC curves to be comparable.

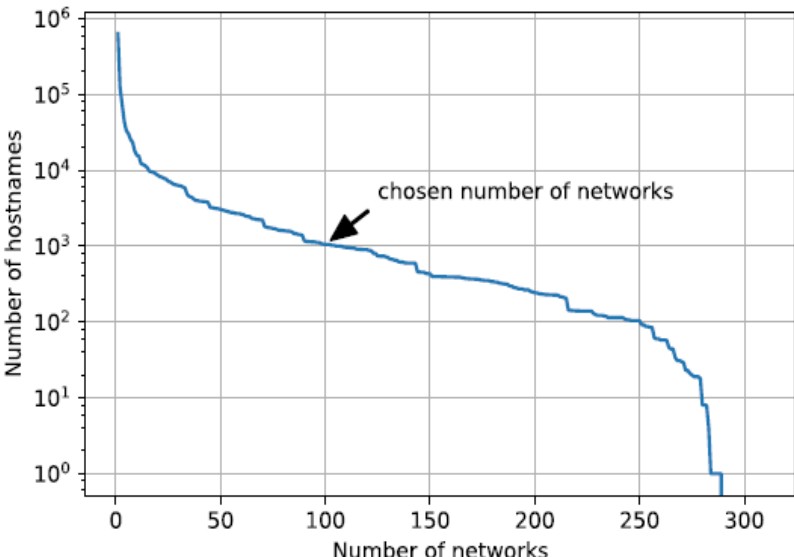

**Figure 6.** Network intersection sizes.

The resulting global reputation score ROC curves are shown in Figure 7. The global reputation table of ROC curve comparison on the biggest 100 networks and the corresponding AUC values are listed in Table 3. The global reputation table of AUC score comparison on the biggest 100 networks shows that the best-performing algorithms are average and pairwise, which are nearly tied in their AUC scores and ROC curve shapes. The average algorithm delivered the best performance and pairwise occupies a close second place. Of the other two algorithms, norm-all is in third place and norm-isect clearly performed worst.

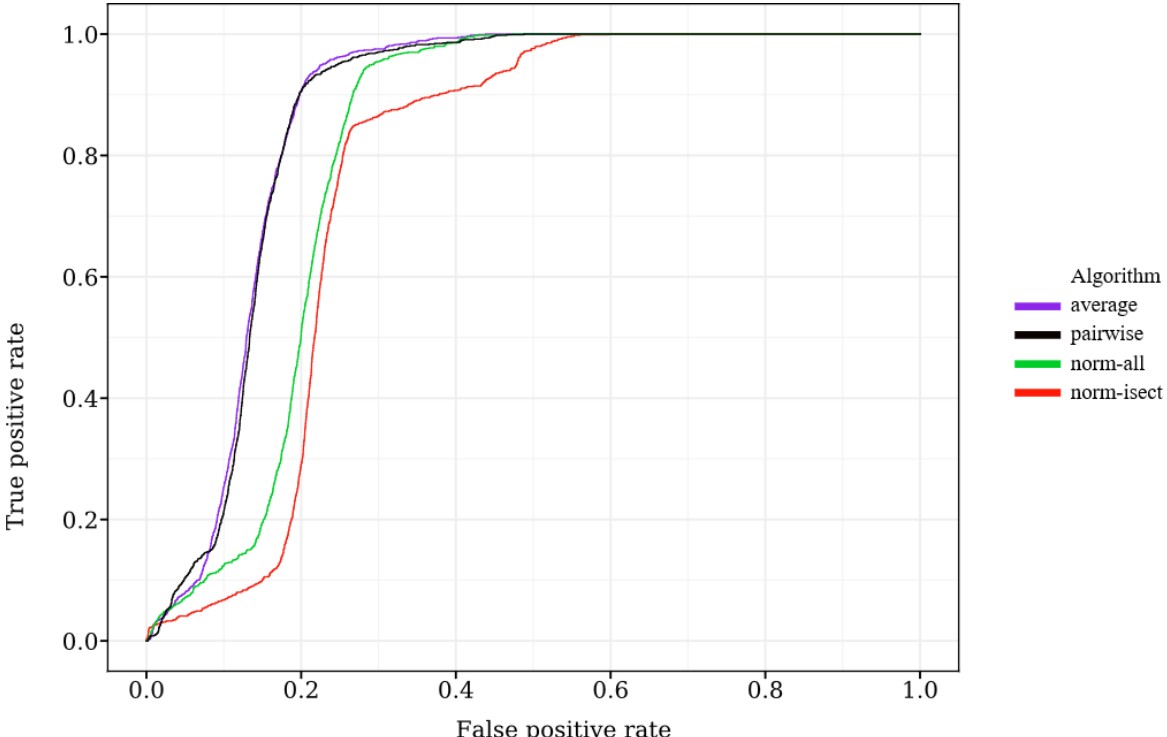

**Figure 7.** Global reputation table receiver operating characteristic (ROC) curve comparison on the biggest 100 networks.

As expected, norm-isect did not perform particularly well. It was able to exploit only a tiny fraction of the available information to perform normalization: only 1054 hostnames were used compared to roughly 6.1 million hostnames utilized by norm-all. This probably caused substantial noise in the estimation of normalization parameters, degrading the ensemble performance. While norm-all exploits all of the available data, it suffers from a different problem: it standardizes every network output in isolation. The sets of observed hostnames are different for different networks, which then translates to different statistical characteristics of the individual network anomaly detection outputs. Even if networks were perfectly normalized (returning identical anomaly scores for each hostname), statistical differences between their outputs would exist and be erroneously corrected by the norm-all algorithm.

The pairwise algorithm utilized 1.2 million hostnames (18% of total) and performed on percentage% with the average algorithm. The pairwise algorithm could have improved upon the average algorithm by performing normalization and thus removing spurious calibration differences. On the other hand, it could have decreased performance by removing meaningful calibration differences, or by making an inaccurate estimate of the normalization parameters due to noisy data. The positive and negative effects have canceled out to leave the performance of pairwise and average nearly equivalent.

As can be observed in the next section, there are benefits to using pairwise normalization over a simple average, but the positive effect manifests mainly for the worst-performing networks. This improvement was likely diluted in the mostly unchanged performance of typical networks and did not manifest in the global reputation table of AUC scores discussed above, as is clear in following results.

**Table 3.** Global reputation table of area under the ROC curve (AUC) score comparison on the biggest 100 networks.

| Algorithm Names | AUC Score |
|:---:|:---:|
| average | 86.6% |
| pairwise | 86.2% |
| norm-all | 80.8% |
| norm-isect | 76.8% |

*5.4. Improvement of Per-Flow Anomanly Scores*

In this section, we explore the possibility of using a global reputation table to improve the anomaly scores in individual networks (the third arrow in Figure 2). The reputation model was used in the manner described in Section 4.3. There are two ways in which this procedure might improve classification.

1. Smoothing: Since many values are averaged together to arrive at the reputation score, much of the noise of the detectors will be filtered out.
2. Information sharing: Many different networks participate in the creation of the reputation table, which is beneficial as discussed in Section 1.

If smoothing is the main benefit of the reputation table, it may be possible to simply build a network-local reputation table and use that—no information sharing is necessary. Because of this, smooth evaluation and information sharing separately is necessary to know whether information sharing is beneficial or not. Three types of anomaly detection algorithms were evaluated:

1. Plain: The anomaly scores corresponding to individual flows; no reputation table is used.
2. Local: Anomaly scores are enhanced by a network-local reputation table. Since this reputation table is built using only one network, there are no normalization, combination or missing data issues, and this reputation table is a simple per-hostname average of anomaly scores.
3. Global: Anomaly scores enhanced by a global reputation table. The global reputation table can be generated by either of the four different algorithms described in Section 5.3. Due to their poor performance, norm-all and norm-isect were not attempted; instead focus was on average and pairwise algorithms. By discarding the norm-isect algorithm, the full set of 310 networks was used. The global and pairwise reputation models were re-computed by using all 310 networks and these updated models were utilized in the following experiments.

The rationale behind evaluating local vs. global is that most of the smoothing effect will be present in both of them; the strength of information sharing (mostly) alone can be assessed by comparing their performance. If the global algorithms significantly improve classification strength over both plain and local algorithms, information sharing is beneficial.

In order to test the effects of information sharing on HTTP and HTTP(S) flows separately, the performance of the algorithms both in isolation and combined will be tested.

This gives us four different classification algorithms over three distinct datasets (per network), for a total of 3720 distinct performance measures. This number of different results makes the ROC curves too granular; instead, only the AUC scores will be used to evaluate the classification strengths. More than half of the networks do not contain even a single flow labeled as anomalous. For these, the AUC score is not meaningful and they are excluded from the comparison, leaving just 1432 performance measures.

We expect the classification performance of "plain" to be significantly worse for HTTP(S) traffic than for HTTP traffic because of the lower information content discussed in Section 2. The reputation table should then improve on HTTP(S) traffic more than on HTTP traffic because of the sharing of information gathered by MITM attacks in some networks.

The performance of all tested algorithms is visualized as a box plot in Figure 8A,B. Numerical comparison of AUC per-flow anomaly score is presented in Table 4. It is visible from both Figure 8 and Table 4 that the largest differences between different algorithms are in the lowest percentiles. The 5th percentile (displayed in the Figure 8 as lower whiskers) is a robust indicator of the behavior of networks with poor classification strength. In this percentile, both global reputation algorithms improve substantially upon the base case of "plain" with "pairwise" offering the best performance. Local also improves the 5th percentile, but not nearly as much as the global algorithms. The median does not differ very considerably or predictably between algorithms with the exception of HTTP(S) traffic, where "local" and "average" give inferior results. Interestingly, the pairwise algorithm did not suffer from this effect nearly as much. Another interesting feature of the results is the distribution of outliers (displayed as circles) as given Figure 8B, which shows that the pairwise algorithm eliminated all outliers with AUC < 88%, but every other algorithm suffered from at least some networks with severely degraded performance. This indicates the high robustness of the pairwise reputation model.

Of all the tested algorithms, "pairwise" exhibited the best AUC mean, minimum, 5th and 10th percentiles, and its standard deviation was smallest for all data sets. Other performance characteristics (including the global reputation table ROC curve) were never substantially worse than for other algorithms. These results suggest that "pairwise" is an effective algorithm for building a global reputation model GRM.

Surprisingly, not much difference is immediately obvious between HTTP and HTTP(S) results despite the very different characteristics of their underlying data. The Wilcoxon signed-rank test was used [41–43] to test the significance of differences between all pairs of HTTP and HTTP(S) measurements: plain, local, global and pairwise.

Figure 8A,B shows that Plot (A) differs from plot (B) only in the scale of the y-axis to highlight different parts of the plots. Boxes span between the 25th and 75th percentiles, whiskers span between the 5th and 95th percentiles, and circles represent all the networks outside the range of the whiskers. The median is marked by a horizontal line.

AUCs are expressed as percentages in Table 4. The highest value in each row is bold with the exception of the standard deviation. However, the differences are very small in some rows (median) while being very large in others (minimum, 5th percentile). Higher percentiles are not displayed because their differences were mostly negligible. The maximum was in all cases equal to 100.0%.

In Figure 9, the top chart contains AUCs computed by the plain algorithm and the bottom chart contains AUCs computed by the pairwise algorithm. In both cases, the AUC scores are not linearly dependent on the amount of network traffic. Networks with both great and poor performance are found in all the parts of the size spectrum.

**Table 4.** Comparison of AUCs of per-flow anomaly score.

| Data | Algorithm | Mean | Standard Deviation | Minimum | 5th Percentile | 10th Percentile | 25th Percentile | Median |
|---|---|---|---|---|---|---|---|---|
| **All (HTTP + HTTP(S))** | Plain (%) | 97.3 | 4.4 | 69.1 | 88.5 | 94 | 97.4 | **98.6** |
| | Local (%) | 97.4 | 4 | 65.2 | 91.3 | 94.3 | 97 | 98.2 |
| | Average (%) | 97.3 | 4.8 | 51.9 | 92.8 | 94.5 | 96.6 | 98.4 |
| | Pairwise (%) | **98.1** | 2 | **88.2** | **94.7** | **96.4** | **97.5** | 98.5 |
| **HTTP Traffic** | Plain (%) | 97.1 | 4.8 | 72.7 | 88.2 | 91.9 | 97 | 98.7 |
| | Local (%) | 97..2 | 6.4 | 33.8 | 91.3 | 94.6 | 96.9 | 98.7 |
| | Average (%) | 98.3 | 2.4 | 85.9 | 94.1 | 95.9 | **98.1** | **99.2** |
| | Pairwise (%) | **98.4** | 2.2 | **86.3** | **94.9** | **96.3** | 98 | 99.1 |
| **HTTP(S) Traffic** | Plain (%) | 97.9 | 3.3 | 74.8 | 92 | 96.1 | **97.9** | **98.9** |
| | Local (%) | 97.8 | 2.1 | 87.6 | 93.9 | 95.5 | 96.9 | 98.3 |
| | Average (%) | 96.3 | 7.8 | 44.1 | 93.7 | 94.3 | 96.5 | 97.9 |
| | Pairwise (%) | 98.1 | 1.3 | **94.3** | **95.2** | **96.2** | 97.5 | 98.3 |

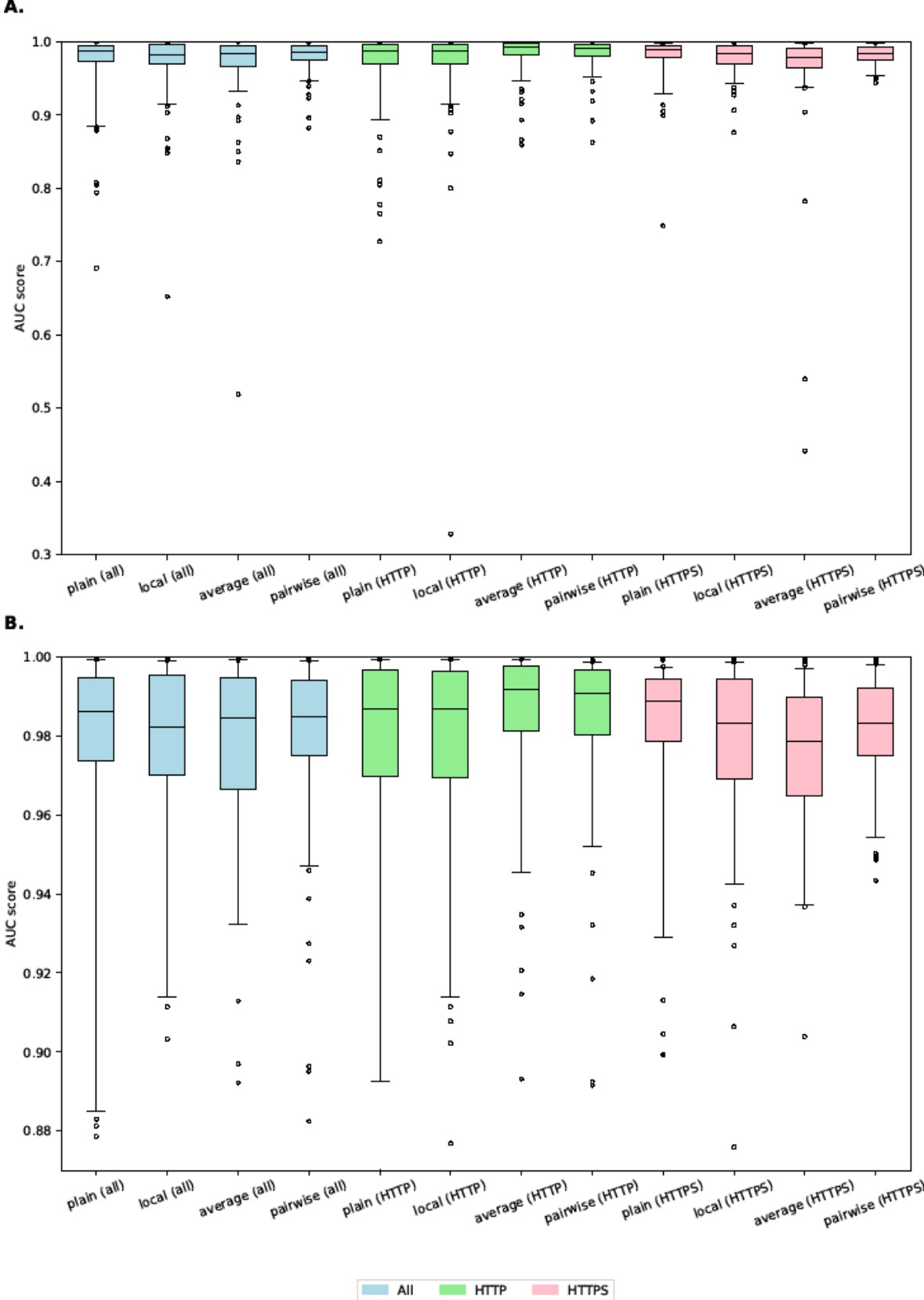

**Figure 8.** Comparison of AUCs (**A**) per-flow anomaly score (**B**) all pairwise algorithm.

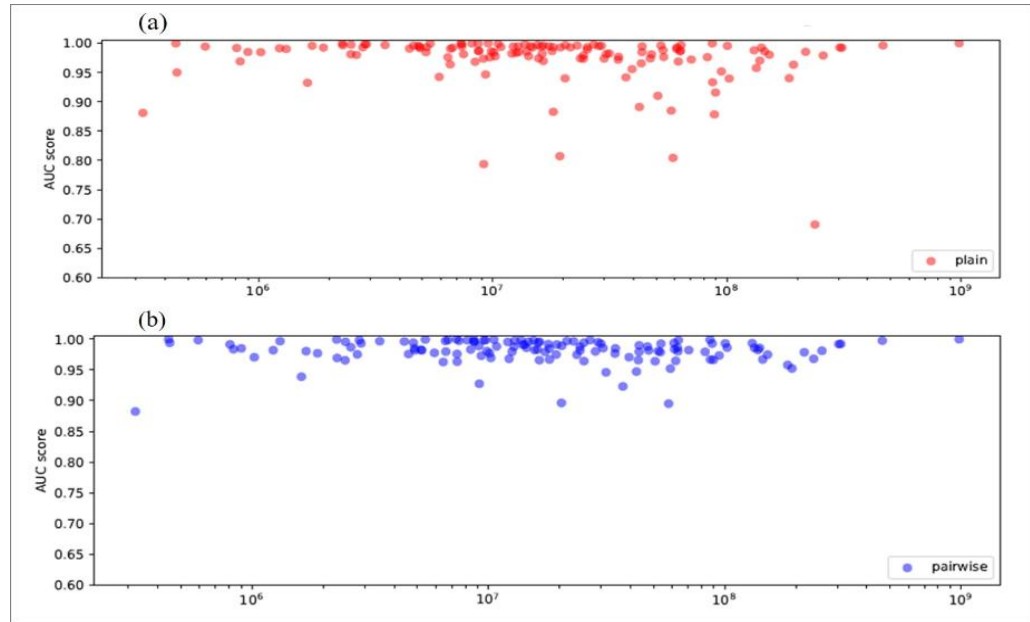

**Figure 9.** Network AUC score and network traffic size on all input data (HTTP and HTTP(S)): (**a**) network traffic size (number of flows) with respect to "plain", (**b**) network traffic size (number of flows) with respect to "pairwise".

For neither of these were there any significant differences between their population mean ranks ($p < 0.01$).

Based on the considerations in Section 1.2, one would think that a primarily small-world network [44] would exhibit a low AUC score: AUCs would be expected to be correlated with the amount of network traffic because small-world networks have less information to train their classifiers on. Surprisingly, this is not the case as illustrated in Figure 9. Similarly, AUC improvement by the reputation tables was not visibly correlated with network size. Even big networks [45] can obviously exhibit bad classification characteristics and benefit from the improvement by global reputation tables.

## 6. Conclusions and Future Work

The main goal of this research work was to improve the detection efficacy of the existing Cognitive Threat Analytics (CTA) anomaly-based network IDS on malware that uses network traffic encryption to avoid detection. Therefore, we propose to utilize the fact that CTA-IDS is used in a large number of various networks and share threat information between them, since the amount and quality of features that can be extracted from HTTP(S) network traffic was low.

The global threat intelligence method was created and shared among all the CTA-IDS systems that are monitoring the individual enterprise networks to improve the anomaly detection efficacy on both encrypted and non-encrypted network traffic. Several variations of the method were developed. One such method incorporates a novel method for outlier ensemble normalization in the presence of large numbers of missing values.

Experimental evaluation performed on a large amount of real-world network traffic showed that the proposed algorithm performs only marginally better than the relevant state-of-the-art methods in the average case, but outperforms all of the compared methods in the worst-case scenario. Additionally, the efficacy improvements were measured on HTTP(S) communication, and the proposed algorithm and detection capability on HTTP network traffic were improved.

Apart from improving detection accuracy in participating networks, the gathered intelligence can also be used on its own to help better understand global statistics of network threat behavior, and to help network analysts assess threat severity, determine the scale of an attack, or analyze command and control mobility patterns.

Future work: There are a large number of possibilities to improve the final accuracy and the false positive rate in future research work. One possibility is to employ a greater amount of training and testing samples, and to find other available information in BRO logs describing HTTP(S) traffic in different ways or to find different open source software describing HTTP(S) traffic. Next, more in-depth studies could be conducted to analyze how malware works in HTTP(S) and also to try more machine learning algorithms with more different parameters.

**Author Contributions:** Conceptualization, M.T., M.L. N.A. and M.A.; Formal analysis, M.T., M.L., N.A. and M.A.; Funding acquisition, M.L.; Methodology, M.T.; Project administration, M.L., M.T., M.A. and N.A.; Software, M.T., M.A. and N.A.; Supervision, M.L.; Validation, M.T.

**Funding:** This work has been supported by National Natural Science Foundation (NSFC) of P.R. China under grant number 61572095 and 61877007.

**Conflicts of Interest:** The authors declare no conflict of interest.

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
