# Peer review of "Efficacy Improvement of Anomaly Detection by Using Intelligence Sharing Scheme"

_applsci, doi:10.3390/app9030364_

Round 1

Reviewer 1 Report

This paper addresses the problem of improving the network flow-based

detection of malicious communication performed by the Cisco Cognitive

Threat Analytics IDS by using the features provided this monitoring

system and aggregating information provided from several instances on

the system across different networks.

While the authors put certain effort into performing a series of

experiments and evaluating their idea of global and individual

detection, the paper presents, unfortunately, a large number

of important flaws.

In particular:

- The English language used is very poor and sometimes broken to the

  point of rendering some sentences very hard to understand. Moreover,

  the are some strange things like the use of parenthesis in sentences

  that do not require them (e.g. lines 91, 94, 203, 205, 207, 212,

  254, 261, 264, 267, 715, 717, etc). In general, the authors need to

  reread the text and be sure that every sentence has a proper

  structure (subject, verb, etc).

- The structure of the text could be largely improved. For instance,

  the authors describe the different types of IDS in the introduction

  but this is unrelated to the topic of the paper. Instead, the

  authors could clearly state the problem they are trying to address

  and explain how their approach improves over existing approaches.

  Also, lines from 54 to 64 could be indented.  Although it can be

  understood from reading the text, it is rather confusing to have all

  bullet points at the same indentation.  Moreover, the challenges faced by

  network anomaly detection would be better in a background section or

  in the introduction than in the related work. Here, the

  authors should review previous literature where machine learning is

  applied to network flows for malware detection. There is plenty of

  it and it and would allow the authors to motivate their work.

  Additionally, the section "Proposed architecture" should not be in

  the related work section.  Again in 210, bullets and indentation

  would help. Finally, the authors mention results from the

  experiments during the explanation of the method, as in line 500

  (eg. "specifically 99.2%"). Such references are out of context and

  should be removed.

- Regarding the methodology and the evaluation, it is clear from the

  results, that the approach does not work properly. In any case, it

  is difficult to make a clear assertion of the performance. Here, I

  see several issues. For instance, in Figure 7, for the TPR to reach

  a reasonable value, the FPR needs to be almost 20%. This is

  extremely large for a detection system. For a 10% FPR, which is

  already very large, the TPR is very low.  In the case of Figure 8,

  the AUC between 0 and 1 only tells us, roughly, how different

  methods compare with each other, but no information about the TPR at

  a low FPR is provided. This is essential to understand how a

  classifier is performing for a specific problem.

OTHER COMMENTS:

- Figure 2 proportions distort the image.

- L15: What does sharing intelligence mean?

- L683: Strange character over the text. 

- Figure 5 should not have lines, it is an histogram, company IDs are

  independent, they should be bars.

TYPOS (only some here, there are many more):

- L16: In *the* second

- L17: variants __ based 

- L728: "Figure 5. Flow counts per network."

- L729 Networks/networks

Author Response

First of all, we would like to thank the reviewer for the excellent suggestions and comments. Without these suggestions, this manuscript would not have been in the current shape and form. Based on the valuable reviews and comments of the reviewers, we have thoroughly revised the manuscript. We hope that the reviewers will find that the current revision is up to their required standards. 

Response to Reviewer 1 Comments

Reviewer #1

This paper addresses the problem of improving the network flow-based detection of malicious communication performed by the Cisco Cognitive Threat Analytics IDS by using the features provided this monitoring system and aggregating information provided from several instances on the system across different networks.

While the authors put certain effort into performing a series of experiments and evaluating their idea of global and individual detection, the paper presents, unfortunately, a large number of important flaws.

Point-1: The English used is very poor and sometimes broken to the point of rendering some sentences very hard to understand?

Response-1: We have revised every section carefully, identify grammar and spelling issues and then rectified errors.

Point-2: Moreover, there are some strange things like the use of parenthesis in sentences that do not require them (e.g. line 91, 94, 203, 205, 207, 212, 254, 261, 264, 267, 715, etc.). In general, the authors need to reread the text and be sure that every sentence has a proper structure (Subject, Verb, etc.).

Response-2: Thanks for the anonymous reviewer for this excellent comment. Yes there were grammatical mistakes in the manuscript. In our updated manuscript we have read each section carefully and the errors are rectified.  We have improved the structure of the sentences throughout the manuscript, now all the sentences are more meaningful and in sequence.

Point-3: The structure of the text could be largely improved. For instance, the authors describe the different types of IDS in the introduction but this is unrelated to the topic of the paper. Instead, the authors could clearly state the problem they are trying to address and explain how their approach improves over existing approaches.

Response-3: Thanks for this valuable comment. According to the suggestions of reviewer, we have added the major challenges (1.1. Major Challenges in Anomaly-based IDS) and main goals (1.2 Main Goals and Contributions) of this study in the Introduction section. 

Point-4: Also, line from 54 to 64 could be indented. Although it can be understood from reading the text, it is rather confusing to have all bullet points at the same indentation. Moreover, the challenges faced by network anomaly detection would be better in a background section or in the introduction than in the related work. Here, the authors should review previous literature where machine learning is applied to network flows for malware detection. There is plenty of it and would allow the authors to motivate their work.

Response-4: This is very good observation by the reviewer, we have rearranged the paragraphs. we have move the section related to the Major Challenges in Anomaly-based IDS in the introduction section. Due to the re-arrangements of the paragraphs, the lines “54-64” are moved to “58-66”.

Point-5: Additionally, the section “Proposed architecture” should not be in the related work section. Again in 210, bullets and indentation would help. Finally, the authors mention results from the experiments during the explanation of the method, as in line 500 (e.g. “specifically 99.2 %”). Such reference are out of context and should be removed.

Response-5: This is also a good observation and thanks for this valuable suggestion. We have moved the section related to the proposed architecture from related work and combined the sections related to the proposed architecture and Proposed Algorithm in Section 4. Additionally, we have removed the reference “specifically 99.2 %” (Line # 450).

Point-6: Regarding the methodology and the evaluation, it is clear from the results that the approach does not work properly. In any case, it is difficult to make a clear assertion of the performance. Here, I see several issues. For instance, in Figure 7, for the TPR to reach a reasonable value, The FPR needs to be almost 20%. This is extremely large for a detection system. For a 10% FPR, which is already very large, the TPR is very low. In the case of Figure 8, the AUC between 0 and 1 only tells us, roughly, how different methods compare with each other, but no information about the TPR at a low FPR is provided. This is essential to understand how a classifier is performing for a specific problem.

Response-6: Thanks for your observation, but according to our experimental evaluation (Section 6.2), we compared TCP flags anomaly detectors with anomaly measures (Section 5.2, Equation #3) (Figure 4). Further, referred both with link aggregation bonding (Fig 1.). over source and destination IP (srcIP, dstIP) addresses (5 detectors in total) with other existing anomaly detectors from the prior art described detail in (section 2). To compare these methods we have created the Receiver Operation Characteristics (ROC) of each of the mentioned anomaly detection methods, chosen several number of real networks (Fig 6.) and number of flow counts per network according to the number of company’s identification ID (Fig 5.). Next, we have evaluated the ROC’s Area Under the Curve (AUC) to be able to easily compare all the presented methods. Because we are interested only in the false positive rates smaller than 1% of all the traffic, we computed the ROC’s AUC as an integral of the ROC up to 1% of false positive threshold. This gives us good insight about the detector’s performance for small false positive threshold, which is important to the field of network anomaly detection.

Other Comments:

·         Figure 2 proportions distort the image.

Response: We have improved the quality of the image (Figure 2).

·         L15: What does sharing intelligence mean?

Response: Intelligence sharing is meant to facilitate the user of an actionable intelligence to a broader range of decision making.

·         L683: Strange character over the text.

Response: This was typo, we have rectified the error.

·         Figure 5 should not have lines, it is a histogram, company IDs are independent, and they should be bars.

Response: In figure 5, we were trying to show the number of companies IDs according to the number of networks. As companies IDs are more than 300, it is difficult to show these ids on Histogram.

TYPOS (only some here, there are many more):

·         L16: In *the* Second

Response: We have rectified the error.

·         L17: variants-based

Response: We have rectified the error.

·         L728: “Figures 5. Flow counts per network.”

Response: We have rectified the error.

·         L729: Networks/networks

Response: Yes, It’s a word “Networks”.

Reviewer 2 Report

The authors propose a scheme to improve anomaly detection using threat intelligence. The paper is well written however, will require a thorough proofreading as numerous grammatical mistakes are made within the manuscript.

Section 2 Related Works describes the challenges of the anomaly detection, while most points made are valid, it is surprising to see that no recent work is being considered within the description of this section. I.e. recent work has been published by Al Tobi and Duncan (2018) on the problems of the KD99 datasets ,  Hindy et al. (2018) published a review on the issues of current datasets and the lack of representative attacks, while line 151, touches this problem, I would advise to revise the section and add the problems described hereabove. 

The authors are not clear on how this scheme is different from collaborative anomaly detection systems. 

Equation 1,2,4 and 5 should be better described and information related to the equations should be highlighted within the text. 

In Section 5.1. the authors describe the datasets used. While the dataset is private, but represent a real life example, could the authors highlight how their experiments can be reproduced by other teams.

Can the authors provide a justification of the split of data, why those splits ? have they been chosen at random or over multiple trial and error, etc. 

ROC and AUC have been critisized by multiple researchers succh as Hanczar et al (2010), Lobo et al. (2008) and Hand (2009). Could the authors provide a justification for its usage. 

The future work highlights some solutions to decrease the fp rate , could the author explain why it hasn't been implemented in the main body of work ?

Author Response

First of all, we would like to thank the reviewer for the excellent suggestions and comments. Without these suggestions, this manuscript would not have been in the current shape and form. Based on the valuable reviews and comments of the reviewers, we have thoroughly revised the manuscript. We hope that the reviewers will find that the current revision is up to their required standards. 

Comments Reviewer 2

The authors propose a scheme to improve anomaly detection using threat intelligence. The paper is well written however, will require a thorough proofreading as numerous grammatical mistakes are made within the manuscript.

Point 1. Section 2 Related Works describes the challenges of the anomaly detection, while most points made are valid, it is surprising to see that no recent work is being considered within the description of this section. I.e. recent work has been published by Al Tobi and Duncan (2018) on the problem of the KD99 datasets, Hindy et al. (2018) published a review on the issues of current datasets and the lack of representative attacks, while line 151, touch this problem, I would advise to revise the section and add the problems described here above. 

Response 1: Thanks for the reviewer for this valuable comments. We have thoroughly revised the section and cited the recent articles published by AI Tobi [16, 17] line 139.

Point-2: The authors are not clear on how this scheme is different from collaborative anomaly detection systems.

Response-2: Thanks for this valuable comment. CTA-IDS is already collaborative device installed globally in enterprise-networks, cs-host flow field is not available in HTTP(S) traffic but the host name can be extracted from HTTP(S) certificate information. We are collaborating on the new updated version of software design of CTA-IDS, implementation, and evaluation of new Machine learning algorithms. By the inter-network data correlations in Cisco CTA-IDS, and collaborative anomaly-based networks IDS on malware that uses network traffic encryption to avoid detection described in (Section 2). Further details can be seen in table 1.

Point.3: Equation 1, 2, 4 and 5 should be better described and information related to the equation should be highlighted within the text.

Response-3: Thanks for your suggestion we have updated and highlighted the text related to the equation 1, 2, 4 and 5.

Point.4: In Section 5.1. The authors describe the datasets used. While the dataset is private, but represent a real life examples, could the authors highlight how their experiments can be reproduced by the other teams? 

Response-4: Thanks for your important question, we agree with you on the datasets are private (indeed). We have represent a several number of real network examples because of the extract observable features in the CTA-IDS software testing requirements, and ideas are new routes. So every researcher can get destination faster. Finally, for the perspective of reproducing new experiments other teams also try to get familiar with decision related classification techniques, prepare and get familiar with the training and testing data sets, propose and implement methods in a chosen programming language and evaluate the methods on real world data set.

Point.5: Can the authors provide a justification of the split of data, why those splits? Have they been chosen at random or over multiple trial and error, etc.?

Response-5: Thanks for your excellent comment, In order to test the deployment scenario described in (Section 5.1), the dataset must be splitted chronologically into two parts. The first part is used to create and test a global reputation table while the second part is used to evaluate the possible improvements of per-flow anomaly scores using a reputation table. Coupled with the fact that there must be a warm-up period in which the anomaly detectors are learning the characteristics of network traffic and do not provide useful output.

Point.6: ROC and AUC have been criticized by multiple researchers such as Hanczar et al (2010), Lobo et al. (2008) and Hand (2009). Could the authors provide a justification for its usage?

Response.6: Thanks for your good question, regarding ROC and AUC criticism. We are agree with you on the criticism according to the articles Hanczar et al (2010), Lobo et al. (2008) and Hand (2009), but according to our experimental evaluation (Section 5.2), we compared TCP flags anomaly detectors with anomaly measures (Section 5.2, Equation #3) denoted by (Figure 4). Furthermore, referred both with link aggregation bonding (Fig 1) over source and destination IP (srcIP, dstIP) addresses (5 detectors in total) with other existing anomaly detectors from the prior art described in detail  (section 2). To compare these methods, we have created the Receiver Operation Characteristics (ROC) of each of the mentioned anomaly detection methods chosen several number of real networks (Fig 6.) and number of flow counts per network according to the number of company’s identification ID (Fig 5.). Next, we have evaluated the ROC’s Area Under the Curve (AUC) to be able to easily compare all the presented methods. Because we are interested only in the false positive rates smaller than 1% of all the traffic, we computed the ROC’s AUC as an integral of the ROC up to 1% of false positive threshold. This gives us good insight about the detector’s performance for small false positive threshold, which is important to the field of network anomaly detection.

Point.7: The future work highlights some solutions to decrease the FP rate, could the author explain why it hasn’t been implemented in the main body of work?

Response-7: Thanks for asking this important question, this future work is also in process, our team is working on this project.

Reviewer 3 Report

In this study, the authors propose two new methods to improve the efficacy of the Cisco Cognitive Threat system. In the first method, the efficacy of CTA is improved by sharing intelligence across many enterprise networks. In the second method, four variants based Global Reputation Model is designed by employing outlier ensemble normalization algorithms in presence of missing data. Intelligence sharing provides additional information to the intrusion detection process, which is much needed particularly for analysis of encrypted traffic with inherently low information content.

This paper has the potential to be accepted, but some important points have to be clarified or fixed before we can proceed and a positive action can be taken.

We here summarize these points:

1.    I think the motivation for the present research would be clearer if the authors could provide a more direct link between the importance of choosing the appropriate methods that have proposed and the case study that have presented. One way to demonstrate this connection would be to a comparison with other similar methods.

2.    A major issue of the paper is the explanation on how the authors have chosen this specific architecture for the proposed method, how it emerged and why the proposed architecture is the optimal solution.  

3.    Also, the authors should clearly define which are the innovative features of their proposal with respect to adopted logics.

4.    We need to understand precisely if the proposed mechanism is able to face other real case scenarios.

5.    Is the design of the study consistent with its aims?

6.    The authors do not include a detailed description of how the method proposed can be extended.

7.    Finally, the authors should make a final check on the compliance of the paper to the formatting guidelines and need to delete the template guidance at the end of the paper.

Author Response

First of all, we would like to thank the reviewers for the excellent suggestions and comments. Without these suggestions, this manuscript would not have been in the current shape and form. Based on the valuable reviews and comments of the reviewers, we have thoroughly revised the manuscript. We hope that the reviewers will find that the current revision is up to their required standards. 

Comments Reviewer #3

In this study, the authors propose two new methods to improve the efficacy of the Cisco Cognitive Threat system. In the first method, the efficacy of CTA is improved by sharing intelligence across many enterprise networks. In the second method, four variants based Global Reputation Model is designed by employing outlier ensemble normalization algorithms in presence of missing data. Intelligence sharing provides additional information to the intrusion detection process, which is much needed particularly for analysis of encrypted traffic with inherently low information content.

This paper has the potential to be accepted, but some important points have to be clarified or fixed before we can proceed and a positive action can be taken.

Point-1. I think the motivation for the present research would be clearer if the authors could provide a more direct link between the importance of choosing the appropriate methods that have proposed and the case study that have presented. One way to demonstrate this connection would be to a comparison with other similar methods.

Response.1: Thanks for this valuable comment. According to the suggestions of reviewer, we have added the major challenges (1.1. Major Challenges in Anomaly-based IDS) and main goals (1.2 Main Goals and Contributions) of this study in the Introduction section, which demonstrate the importance of the proposed method. The comparison of the proposed method with state-of-the-Art method is given in the section 3 (Related work and State-of-the-Art).

Point-2.  A major issue of the paper is the explanation on how the authors have chosen this specific architecture for the proposed method, how it emerged and why the proposed architecture is the optimal solution.

Response.2: CTA-IDS is already collaborative device installed globally in enterprise-networks, there is a gap of cs-host flow field that is not available in HTTP(S) traffic but the host name can be extracted from HTTP(S) certificate information. We are collaborating on the new updated version of software design of CTA-IDS, implementation, and evaluation of new Machine learning algorithms. By the inter-network data correlations in Cisco CTA-IDS, and collaborative anomaly-based networks IDS on malware that uses network traffic encryption to avoid detection described in (Section 2). Further details can be seen in table 1.

Point.3: Also, the authors should clearly define which are the innovative features of their proposal with respect to adopted logics.

Response-3: This study describes the State-of-the-Art algorithms for statically anomaly detection and outlier detection, with emphases on the algorithm used in the network security domain. We have Built a several variants-based Global Reputation Model (GRM) on top of the CTA-IDS, Evaluated the performance of the system variants when used as global reputation lists and Used the global intelligence to improve upon existing anomaly detection capabilities in individual networks and measure the results on both HTTP and HTTP(S) traffic (section 1 “Introduction”, subsection 1.2).

Point-4.    We need to understand precisely if the proposed mechanism is able to face other real case scenarios.

Response-4: Experimental evaluations shows that our proposed algorithm remains consistent and is able to be used in the real time cases scenarios.

Point-5.    Is the design of the study consistent with its aims?

Response-5: Yes, experimental evaluations shows that our proposed algorithm remains consistent and scalable with its aims.

Point-6.    The authors do not include a detailed description of how the method proposed can be extended.

Response-6: Thanks for asking this important question, this future work is also in process, our team is working on this project.

Point-7. Finally, the authors should make a final check on the compliance of the paper to the formatting guidelines and need to delete the template guidance at the end of the paper.

Response-7: Thanks for this comment, we have removed the lines related to the guidance of the temple from the end of manuscript.

Round 2

Reviewer 2 Report

The authors have addressed all comments. I am happy for the paper to be published.